# Association of ideal cardiovascular health metrics with incident low estimated glomerular filtration rate: More than a decade follow-up in the Tehran Lipid and Glucose Study (TLGS)

**Fatemeh Alizadeh[1], Maryam Tohidi[1]\*, Mitra Hasheminia[1], Firoozeh Hosseini-Esfahani[2], Fereidoun Azizi[3], Farzad Hadaegh[1]**

1 Prevention of Metabolic Disorders Research Center, Research Institute for Endocrine Sciences, Shahid Beheshti University of Medical Sciences, Tehran, Iran, 2 Nutrition and Endocrine Research Center, Research Institute for Endocrine Sciences, Shahid Beheshti University of Medical Sciences, Tehran, Iran, 3 Endocrine Research Center, Research Institute for Endocrine Sciences, Shahid Beheshti University of Medical Sciences, Tehran, Iran

\* tohidi@endocrine.ac.ir

## Abstract

### Aims

To evaluate the association between ideal cardiovascular health metrics (ICVHM) and incident low estimated glomerular filtration rate (eGFR) among the Iranian population.

### Methods

The study population included 6927 Iranian adults aged 20–65 years (2942 male) without prevalent low eGFR [i.e., eGFR < 60 ml/min/1.73 m$^2$] and free of cardiovascular disease. The ICVHM was defined according to the 2010 American Heart Association. The multivariable Cox proportional hazards regression analysis was used to calculate the hazard ratios (HRs) of ICVHM both as continuous and categorical variables.

### Results

Over the median of 12.1 years of follow-up, we found 1259 incident cases of low eGFR among the study population. In this population, ideal and intermediate categories of body mass index (BMI) and blood pressure (BP) and only the ideal category of fasting plasma glucose (FPG) significantly decreased the risk of developing low eGFR; the corresponding HRs and (95% confidence intervals) were (0.87, 0.77–0.99), (0.84, 0.76–0.99), (0.79, 0.68–0.93), (0.70, 0.60–0.83) and (0.76, 0.64–0.91). Also, one additional ICVHM was associated with a reduced risk of low eGFR for the global (0.92, 0.88–0.97) and biological cardiovascular health (0.88, 0.82–0.93) in these participants. A sensitivity analysis using the interval-censoring approach demonstrated that our method is robust, and results remained essentially unchanged. In a subgroup population with dietary data (n = 2285), we did not find the

**Data Availability Statement:** The data underlying the results presented in the study are available from SBMU Research Institute for Endocrine Sciences. Furthermore, we uploaded an Excel file containing all data of this study as a "Supporting Information data file".

**Funding:** This study was supported by Grant No. 121 and No. 30382 from the National Research Council of the Islamic Republic of Iran and Shahid Beheshti University of Medical Sciences. The funders had no role in study design, data collection and analysis, decision to publish, or preparation of the manuscript.

**Competing interests:** The authors have declared that no competing interests exist.

beneficial impact of having intermediate/ideal categories of nutrition status compared to its poor one on incident low eGFR.

## Conclusion

We found a strong inverse association between having higher global ICVHM with incident low eGFR among the non-elderly Iranian population; the issue is mainly attributable to normal BP, BMI, and FPG levels.

## Introduction

The prevalence of non-communicable diseases (NCDs), mainly cardiovascular diseases (CVD), cancers, hypertension, and diabetes, has been rising continuously over the past decades. NCDs, also known as chronic diseases, tend to be of long duration and are the result of a combination of genetic, physiological, environmental and behavioral factors [1], mainly due to the enhancement of technology, improvement of life expectancy, and the aging trend of the population [2, 3]. In 2016, it was estimated that NCDs were the leading cause of death, comprising more than 70% of total global mortality [4]. Among NCDs, chronic kidney disease (CKD) is a prevalent disease associated with CVD. The prevalence of CKD has been estimated to be approximately 8 to 16% globally and has continuously increased in the past few years [5, 6]. In Iran, based on the meta-analysis conducted in 2018, the prevalence of CKD was approximately 15.14% [7]. Moreover, more than 2% of Tehranian adults each year develop CKD [8]. Hosseinpanah et al. reported the prevalence of CKD to be more than 45% in Tehranian people over 60 years of age [9].

Diabetes mellitus (DM), high blood pressure (BP), hypercholesterolemia, high body mass index (BMI), and smoking are common risk factors for CKD and CVD [10]. Common pathophysiology of cardiovascular and renal diseases suggests that preventing and properly managing cardiovascular risk factors may also effectively prevent CKD [11]. In 2010, American Heart Association (AHA) established an initiative called Life's Simple 7 to improve cardiovascular health among the American population. It is comprised of seven common risk factors for CVDs, including; smoking, BMI, physical activity (PA), diet, BP, total cholesterol (TC), and fasting plasma glucose (FPG). Since then, several studies have shown that achieving more ideal cardiovascular health metrics (ICVHM) are also associated with a lower incidence of type 2 DM (T2DM) and CVD and better cognitive function and quality of life [12–18]. Only a limited number of studies conducted among the adult population of the United States and in the East Asian population evaluated the association between ICVHM and the incidence of CKD and reported that higher ICVHM significantly reduced the risk of CKD and end-stage renal disease (ESRD) [11, 19, 20].

According to the national data, the prevalence of ICVHM remained very low from 2007–2016, and in the last survey, less than 4% of Iranian adults met $\geq 6$ criteria of ICVHM. Although the smoking, BP status, and TC level had improved slightly, the unfavorable trends of PA, unhealthy diet, obesity, and dysglycemia were reported [21]. In the current study, for the first time, we examined the association of ICVHM and incident low estimated glomerular filtration rate (eGFR), i.e., less than $< 60$ ml/min/1.73 m$^2$ in a large population-based study conducted in the Middle East and North Africa (MENA), a region with a high burden of CVD risk factors [22] during more than 12 years of follow-up.

## Methods

### Study population

The ethics statement was issued by Shahid Beheshti Medical University with approval number: IR.SBMU.ENDOCRINE.REC.1400.139. The consent was obtained both written and oral from each participant.

The Tehran Lipid and Glucose Study (TLGS) is a prospective longitudinal cohort study that was started in 1999 in Tehran, the most populated city in Iran. The purpose of this study was to evaluate the prevalence and incidence of NCDs and their risk factors and to promote healthier lifestyles through education. The study population was selected through random multi-stage cluster sampling, and the time interval between follow-up visits was about three years. (phase 1: 1999–2002, phase 2: 2002–2005, phase 3: 2005–2008, phase 4: 2008–2011, phase 5: 2011–2014, phase 6: 2014–2018, phase 7: 2018–2021) Detailed information regarding the TLGS is available elsewhere (trial registration number: ISRCTN52588395) [23–25].

In the current study, 8624 subjects 20–65 years old who participated in the TLGS from 2005 to 2008 were enrolled. After excluding those with baseline eGFR of less than 60 ml/min per 1.73 m$^2$ (n = 356), history of CVD (n = 402), missing data (n = 548), or those without any follow-up after baseline enrollment (n = 391), 6927 subjects who followed till February 2021 remained for the analyses. Of these participants, 2419 individuals had dietary data. After excluding subjects with under- and over-reporting of energy intake ($< 800$ or $\geq 4200$ Kcal/day), 2285 individuals remained for subgroup analysis (Fig 1).

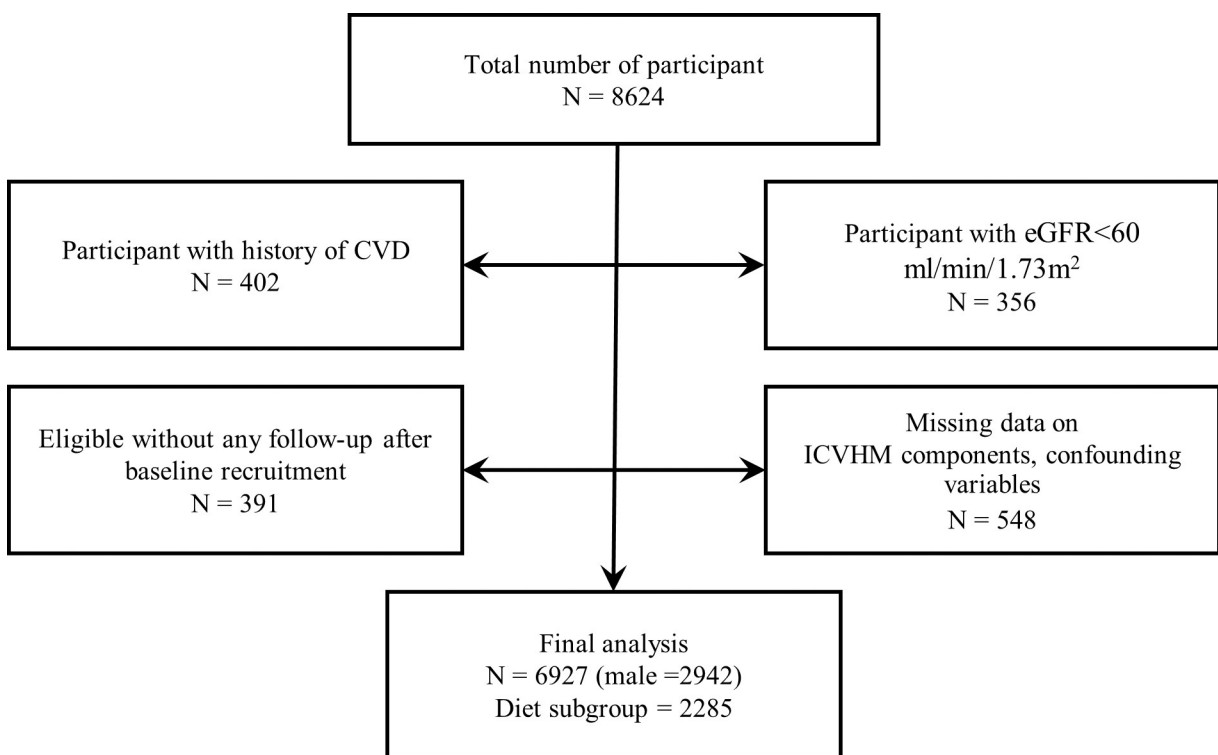

**Fig 1. Fellow chart of study population.** CVD, cardiovascular disease; eGFR, estimated glomerular filtration rate; ICVHM ideal cardiovascular health metrics.

## Clinical and laboratory measurements

A standardized questionnaire was used to gather demographic data and relevant information, such as the history of smoking, medications, and CVD. Also, the modifiable activity questionnaire (MAQ) [26] and food frequency questionnaire (FFQ) [27, 28] were also set for gathering data regarding PA and nutrition, respectively. The Persian version of MAQ and FFQ was validated in previous studies [26, 28, 29].

A trained nurse performed the anthropometric examinations. Weight was measured with minimal clothing using a calibrated digital weighing scale with an accuracy of 100 grams (Seca 707, range 0.1–150 kg; Hanover, MD, USA). Height was measured in a normal upright position with shoulders at rest and without shoes. BMI was calculated by dividing weight by the square of height (kg/m$^2$). Twice measurements of systolic and diastolic blood pressures (SBP and DBP, respectively) were done by a calibrated sphygmomanometer for each participant after 15 minutes of rest by a trained physician and the averages of those measurements were recorded as BP. Further information is available elsewhere [23, 24].

To gather dietary data, we used a valid and reliable 168-item semi-quantitative FFQ to assess usual dietary intakes over 12 months before the follow-up examination [27, 28]. The consumption frequencies of each food item were converted to daily intakes (in grams) and then were changed to servings per day.

The participants were referred to the laboratory between 7 to 9 AM after 12 to 14 hours of fasting for blood sampling. Samples were centrifuged for 30–45 minutes, and plasma and serum samples were extracted according to the standard protocols. FPG, TC, and creatinine measurements were performed on the same day of blood collection. FPG and serum TC were measured using enzymatic colorimetric methods with glucose oxidase, and cholesterol oxidase and cholesterol esterase, respectively. Serum creatinine (cr) levels were assayed by the kinetic colorimetric Jaffe method with an assay sensitivity of 0.2 mg/dL. All biochemical assays were performed using commercial kits (Pars Azmoon Inc., Tehran, Iran) by a Selectra 2 auto-analyzer (Vital Scientific, Spankeren, The Netherlands). Assay quality was monitored after every 25 tests using lyophilized serum controls in normal and different abnormal concentrations. Intra- and inter-assay coefficients of variations are 2.2% and 2.2% for serum glucose, 2.0 and 0.5% for TC, and 1.2 and 2.6% for creatinine, respectively. The CKD Epidemiology Collaboration (CKD-EPI) formula $eGFR = 141 \times \min(Scr/\kappa, 1)^{\alpha} \times \max(Scr/\kappa, 1)^{-1.209} \times 0.993^{Age} \times 1.018 [if\ female] \times 1.159\ [if\ black]$ was used to estimate eGFR [23, 24]. Since the CKD-EPI equation is expressed only for standardized creatinine values, we reduced the creatinine levels by 5% [30].

## Definitions

We used the AHA 2020 criteria, including ideal, intermediate, and poor status for ICVHM. Behavioral factors include PA, smoking and dietary intake. According to data derived from MAQ, the extent of PA was classified as 1) strenuous activity as more than 1500 metabolic equivalent of task (METS) minutes per week, 2) moderate activity as between 600 to 1500 METS minutes per week, and 3) mild activity as less than 600 METS minutes per week. To match the smoking status to ICVHM, three categories were defined: 1) ideal state that includes never smokers 2) intermediate state that includes former smokers, persons who smoked previously up to one year ago, and 3) poor state that consists of current smokers, persons who currently smoke cigarettes, hookah, or pipe. To define the ideal cardiovascular health profile regarding dietary habits, the diet was classified into five items: fruit and vegetable intake of ≥ 4/5 cup per day, eating seafood ≥ 3.5-ounce serving twice per week, eating three ounces of whole grain per day, sodium consumption of less than 1.5 gram per day, and drinking less

than 36 ounces of sugar-sweetened drinks per week. Subsequently, the participant's dietary status was categorized into 1) ideal (5–6 items), 2) moderate (2–4 items), and 3) poor (0–1 item). BMI $\leq$ 25 kg/m$^2$, between 25 to 29.9 kg/m$^2$, and $\geq$ 30 kg/m$^2$ were considered as ideal, intermediate, and poor categories, respectively.

Biological factors include BP status and FPG and TC levels. Ideal, intermediate, and poor BP status was defined as SBP/DBP $\leq$ 120/80 mm/Hg, 120/80 to 139/89 or taking medication, and $\geq$ 140/90, respectively. Three ranges of FPG as FPG $\leq$ 100 mg/dL, between 100 to 125 mg/dL or treated, and $\geq$ 126 mg/dL were considered as ideal, intermediate, and poor status. Regarding TC, TC levels $\leq$ 200 mg/dL, between 200 to 239 mg/dL or taking medication, and $\geq$ 240 mg/dL were defined as ideal, intermediate, and poor conditions, respectively.

ICVHM was calculated by summing the score of each of its six or seven ICVHM in total and subgroup populations, respectively. The sum of each metrics score would define the individual's total ICVHM score (ranging from 0 to 6). Total ICVHM scores < 2 were considered poor, between 2 to 4 as intermediate, and $\geq$ 5 as the ideal ICVHM score.

## Statistical analysis

Continuous variables were expressed as mean ± standard (SD) deviation or median [inter-quartile range (IQR)] and categorical variables as the frequency (%). The ANOVA, Kruskal-Wallis test, and the Chi-squared tests were used to compare continuous and categorical variables among ICVHM categories. Student t-test and Chi-squared tests were used to compare respondents with non-respondents [those with missing data at baseline, or without follow-up (n = 939)].

Due to lack of interaction between gender and all of the variables (P > 0.2), we performed our analysis in a pooled population. The Multivariate Cox Proportional Hazard models were used to calculate the hazard ratios (HRs) of each of the ICVHM metrics for low eGFR, considering the poor status as the reference in model 1 adjusted for age and gender; model 2: model 1 plus educational level and marital status; and model 3: model 2 plus eGFR baseline. Time to event was defined as the time of censoring or date of incidence of low eGFR, whichever occurred first. The event date for the incident cases of low eGFR was defined as mid-time between the date of the follow-up visit at which the low eGFR was determined for the first time and the most recent follow-up visit before the diagnosis. For censored subjects, the time was the interval between the first and the last observation dates. Study participants who died, lost the follow-up, or did not have low eGFR until the end of the study were considered as censored subjects. The proportional hazards assumption in the Cox model was assessed with the Schoenfeld residual test and log-log survival plots. All P values for proportionality assumptions were > 0.1 in different multivariable-adjusted models (S1 and S2 Tables in S1 File). As a sensitivity analysis, we re- run the analysis on a sub-population with available dietary data.

We performed several sensitivity analyses to test the robustness of our findings. First, to reduce selection bias [31], propensity score (PS), the estimated probability that a participant would have followed in the study, was computed using maximum likelihood logistic regression analysis among the study population. For this reason, the entire baseline measures, including age, sex, BMI, SBP, DBP, FPG, TC, eGFR, education level, PA, smoking, marital status, drug consumption for diabetes, lipid and hypertension, were included in a logistic model as exposures with participation in the follow-up as the outcome; the probability of participation in follow-up was then estimated for every participant. Hence we added PS to the Cox models as another covariate (S3-S5 Tables in S1 File). Second, we re-performed our data analyses using Cox regression analysis with interval-censoring (S6-S8 Tables in S1 File). Third, we treated covariates, including age and marital status as time-varying confounders in our data analysis (S9-S11 Tables in S1 File).

**Table 1. Baseline characteristics of the study participants by categories of global cardiovascular health status* (n = 6927).**

| Variables | Global cardiovascular health status | | |
|---|---|---|---|
| | Poor (n = 1370) | Intermediate (n = 3532) | Ideal (n = 2025) |
| **Continuous variables** | | | |
| Age (year) | 46.4 ± 10.3 | 39.7 ± 11.3 | 31.8 ± 9.68 |
| BMI (kg/m$^2$) | 30.2 ± 4.43 | 28.0 ± 4.40 | 23.9 ± 3.91 |
| SBP (mmHg) | 124 ± 15.7 | 111 ± 14.5 | 102 ± 10.3 |
| DBP (mmHg) | 81.0 ± 9.35 | 73.5 ± 9.53 | 66.6 ± 7.77 |
| FPG (mmol/L) | 5.55 (4.88–6.33) | 4.88 (4.61–5.21) | 4.66 (4.44–4.94) |
| TC (mmol/L) | 5.62 ± 0.98 | 4.88 ± 0.95 | 4.21 ± 0.66 |
| eGFR (ml/min/1.73 m$^2$) | 80.8 ± 13.0 | 84.2 ± 13.1 | 89.2 ± 13.6 |
| **Categorical variables** | | | |
| Gender (male) | 767 (26.1) | 1573 (53.5) | 602 (20.5) |
| Educational level (year) | | | |
| < 6 | 392 (34.4) | 623 (54.7) | 124 (10.9) |
| 6–12 | 733 (17.3) | 2147 (50.8) | 1350 (31.9) |
| > 12 | 245 (15.7) | 762 (48.9) | 551 (35.4) |
| Physical activity (METS) | | | |
| < 600 | 714 (27.8) | 1367 (53.3) | 486 (18.9) |
| 600–1500 | 396 (26.1) | 838 (55.3) | 282 (18.6) |
| > 1500 | 260 (9.1) | 1327 (46.7) | 1257 (44.2) |
| Smoking | | | |
| Current | 306 (36.5) | 466 (55.6) | 66 (7.9) |
| Past | 249 (48.8) | 233 (45.7) | 28 (5.5) |
| Never | 815 (14.6) | 2833 (50.8) | 1931 (34.6) |
| Marital status | | | |
| Married | 1200 (22.3) | 2824 (52.5) | 1360 (25.3) |
| Widowed + Divorced | 85 (32.1) | 139 (52.5) | 41 (15.5) |
| Single | 85 (6.7) | 569 (44.5) | 624 (48.8) |
| Glucose lowering drug use, yes (%) | 279 (66.4) | 133 (31.7) | 8 (1.9) |
| Anti-hypertensive drug use, yes (%) | 87 (53.7) | 73 (45.1) | 2 (1.2) |
| Lipid-lowering drug use, yes (%) | 105 (54.4) | 87 (45.1) | 1 (0.5) |

Values are mean ± SD, median (inter-quartile rang), or n (%).

* Defined according to the number of ideal cardiovascular health metrics: 0 to 1 (poor), 2 to 4 (intermediate), and 5 to 6 (ideal).

BMI, body mass index; SBP, systolic blood pressure; DBP, diastolic blood pressure; FPG, fasting plasma glucose; TC, total cholesterol; eGFR, estimated glomerular filtration rate; METS, metabolic equivalent of task.

Statistical analyses were performed using IBM SPSS Statistics for Windows, Version 22 (IBM Corp), Stata Version 14.0 (StataCorp LLC, TX, USA) and R version 4.1.2. P-values $\leq 0.05$ were considered statistically significant.

## Results

The total number of participants was 6927 (male = 2942). As shown in Table 1, the mean ± SD values for age, BMI and eGFR were 38.7±11.8 years, 27.2±4.28 kg/m$^2$, and 85.0±13.6 ml/min/1.73 m$^2$, respectively. S12 Table in S1 File shows the baseline characteristics of the respondent and non-respondent individuals. Compared to the respondent, the non-respondent persons were younger and had a lower mean BMI and higher values of FPG and eGFR. They were more likely than respondent subjects to be smokers and single.

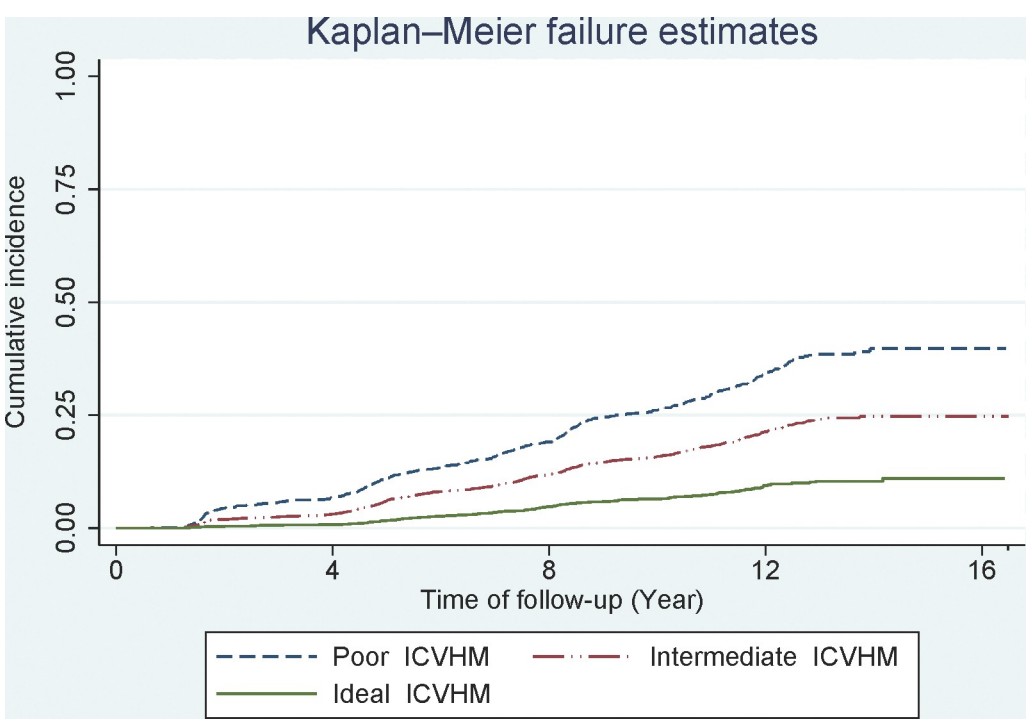

**Fig 2. Kaplan-Mayer curve for poor, intermediate and ideal categories of cardiovascular health and incident developing low estimated glomerular filtration rate.**

The frequencies of different categories of ICVHM were 19.7, 51.0, and 29.3% for poor, intermediate, and ideal categories. There were statistically significant differences in the baseline characteristics between the three categories of ICVHM among participants.

During a median follow-up of 12.1 (9.7–13.2) years, we found 1259 incident cases of low eGFR among the participants. The Kaplan-Mayer curves of poor, intermediate and ideal categories of ICVHM for incident low eGFR are presented in Fig 2. The HRs and 95% confidence intervals (CIs) of intermediate and ideal categories for each ICVHM metric for incident low eGFR are shown in Table 2. According to the results, ideal and intermediate categories of BMI and BP and only the ideal category of FPG significantly decreased the risk of developing low eGFR; the corresponding values were (0.87, 0.77–0.99), (0.84, 0,76–0.99), (0.79, 0.68–0.93), (0.70, 0.60–0.83) and (0.76, 0,64–0.91) in model 3.

As shown in Table 3, one additional ICVHM was associated with (0.92, 0.88–0.97) and (0.88, 0.82–0.93) reduced risk of low eGFR for the global and biological cardiovascular health, respectively, in model 3.

The association between global cardiovascular health status and incident low eGFR is shown in Table 4; intermediate and ideal cardiovascular health categories were demonstrated (0.89, 0.78–1.01, p = 0.06) and (0.71, 0.59–0.86) reduced risk of low eGFR, respectively in model 3. Age, female gender, and lower eGFR were significantly associated with an increased risk of developing low eGFR ($< 60$ ml/min/1.73 m$^2$).

Among 2285 individuals with dietary data, having intermediate/ideal categories of nutrition status compared to poor one was associated with lower risk of incident a low eGFR that did not get rich to the significant level (0.96, 0.88–1.04). Moreover, achieving higher ICVHM was not associated with incident low eGFR, whether as continuous or categorical variables, even in model 1 (S13-S15 Tables in S1 File).

**Table 2. HRs and 95% CIs of intermediate and ideal categories of each ICVHM for incident low eGFR.**

| | | Model 1 | Model 2 | Model 3 |
|---|---|---|---|---|
| | e/N | HR (95% CI) | HR (95% CI) | HR (95% CI) |
| **Smoking status** | | | | |
| Poor | 113/838 | 1 | 1 | 1 |
| Intermediate | 89/510 | 0.88 (0.67–1.16) | 0.88 (0.67–1.17) | 0.87 (0.66–1.15) |
| Ideal | 1057/5579 | 0.95 (0.77–1.17) | 0.96 (0.78–1.18) | 0.94 (0.76–1.16) |
| **Body mass index** | | | | |
| Poor | 463/1732 | 1 | 1 | 1 |
| Intermediate | 559/2883 | 0.91 (0.80–1.03) | 0.80 (0.70–1.01) | **0.87 (0.77–0.99)** |
| Ideal | 237/2312 | **0.84 (072–0.99)** | **0.83 (0.71–0.98)** | **0.84 (0.71–0.99)** |
| **Physical activity** | | | | |
| Poor | 402/2567 | 1 | 1 | 1 |
| Intermediate | 297/1516 | 1.01 (0.87–1.18) | 1.00 (0.86–1.16) | 1.03 (0.88–1.19) |
| Ideal | 560/2844 | 1.04 (0.91–1.18) | 1.03 (0.91–1.17) | 1.04 (0.92–1.18) |
| **Total cholesterol** | | | | |
| Poor | 207/644 | 1 | 1 | 1 |
| Intermediate | 470/1778 | 0.99 (0.84–1.17) | 0.98 (0.83–1.16) | 0.98 (0.83–1.16) |
| Ideal | 582/4505 | 0.86 (0.73–1.02) | **0.85 (0.72–1.00)*** | 0.88 (0.74–1.03) |
| **Blood pressure** | | | | |
| Poor | 236/620 | 1 | 1 | 1 |
| Intermediate | 455/1975 | **0.79 (0.68–0.93)** | **0.79 (0.67–0.93)** | **0.79 (0.68–0.93)** |
| Ideal | 568/4332 | **0.72 (0.61–0.84)** | **0.71 (0.60–0.84)** | **0.70 (0.60–0.83)** |
| **Fasting blood glucose** | | | | |
| Poor | 144/354 | 1 | 1 | 1 |
| Intermediate | 232/761 | 0.96 (0.78–1.19) | 0.95 (0.77–1.17) | 0.84 (0.68–1.04) |
| Ideal | 883/5812 | **0.83 (0.69–0.99)** | **0.82 (0.68–0.98)** | **0.76 (0.64–0.91)** |

HR, hazard ratio; CI, confidence interval; ICVHM: ideal cardiovascular health metrics; eGFR: estimated glomerular filtration rate; e/N, number of low eGFR events /number of population at risk.

Model 1: adjusted for gender and age; Model 2: further adjusted for educational level and marital status; Model 3: further adjusted for eGFR.

Significant values are bold (P value < 0.05)

*P value = 0.06

We performed three sensitivity analyses. First, after further adjustment for PS, our results remained essentially unchanged, as shown in S3-S5 Tables in S1 File. The PS was not associated with incident eGFR < 60 ml/min/1.73m$^2$ in different adjusted models, hence, the

**Table 3. Cox proportional hazard models of ICVH for incident low eGFR (per one additional metric).**

| | Model 1 | Model 2 | Model 3 |
|---|---|---|---|
| | HR (95% CI) | HR (95% CI) | HR (95% CI) |
| Global cardiovascular health | **0.93 (0.88–0.97)** | **0.92 (0.88–0.96)** | **0.92 (0.88–0.97)** |
| Behavioral cardiovascular health | 0.98 (0.91–1.06) | 0.98 (0.91–1.06) | 0.99 (0.91–1.07) |
| Biological cardiovascular health | **0.88 (0.82–0.93)** | **0.87 (0.82–0.93)** | **0.88 (0.82–0.93)** |

ICVH: ideal cardiovascular health; eGFR: estimated glomerular filtration rate; HR, hazard ratio; CI, confidence interval.

Model 1: adjusted for gender and age; Model 2: further adjusted for educational level and marital status; Model 3: further adjusted for eGFR.

**Table 4. HRs and 95% CIs of different global cardiovascular health status † for incident low eGFR.**

|  |  | Model 1 | Model 2 | Model 3 |
|---|---|---|---|---|
|  |  | HR (95% CI) | HR (95% CI) | HR (95% CI) |
| Global Cardiovascular health |  |  |  |  |
|  | Poor | 1 | 1 | 1 |
|  | Intermediate | 0.90 (0.79–1.02) | **0.89 (0.78–1.00) *** | **0.89 (0.78–1.01)*** |
|  | Ideal | **0.72 (0.59–0.87)** | **0.70 (0.57–0.85)** | **0.71 (0.59–0.86)** |
| Age, year |  | **1.11 (1.10–1.12)** | **1.12 (1.11–1.128)** | **1.08 (1.07–1.09)** |
| Female (male as reference) |  | **1.92 (1.70–2.17)** | **2.07 (1.82–2.35)** | **1.56 (1.37–1.78)** |
| Educational level |  |  |  |  |
|  | > 12 | __________ | 1 | 1 |
|  | 6–12 | __________ | 1.06 (088–1.26) | 1.17 (098–1.40) |
|  | < 6 | __________ | **0.80 (0.66–0.98)** | 0.95 (0.78–1.16) |
| Marital status |  |  |  |  |
|  | Married | __________ | 1 | 1 |
|  | Divorced + Widowed | __________ | 0.87 (0.71–1.06) | 0.93 (0.76–1.14) |
|  | Single | __________ | 0.91 (0.66–1.26) | 0.95 (0.69–1.32) |
| eGFR |  | __________ | __________ | **0.93 (0.93–0.94)** |

HR, Hazard Ratio; CI, Confidence Intervals; eGFR, estimated glomerular filtration rate.

† Defined according to the number of ideal metrics: 0 to 1 (poor), 2 to 4 (intermediate), and 5 to 6 (ideal).

Model 1: adjusted for gender and age; Model 2: further adjusted for educational level marital status; Model 3: further adjusted for estimated glomerular filtration rate.

* P = 0.06

selection bias probably did not affect our estimations. Second, considering the interval-censoring approach, our results remained essentially unchanged, as demonstrated in S6-S8 Tables in S1 File. Finally, treating age and marital status as time-varying confounders did not affect our estimates in the data analysis (S9-S11 Tables in S1 File).

## Discussion

This prospective cohort study evaluated the association between ICVHM and incident low eGFR in a long-term population-based study conducted in the MENA region. During a median follow-up of about 12 years among cardiovascular health metrics in non-elderly Tehranian participants, the intermediate and ideal categories of BMI and BP, and ideal category of FPG were associated with reduced risk of incident low eGFR. Moreover, we found that the risk of low eGFR decreased by 8% (HR: 0.92) for each additional global ICVHM after adjustment for age, gender, educational level, marital status, and eGFR. Being in the ideal category of global ICVHM was also associated with about 30% lower risk of incident low eGFR (HR: 0.71); moreover, a signal of lower risk was also found for intermediate ICVHM, as well.

To our knowledge, only three prospective studies conducted among the general population examined the association between ICVHM and kidney impairment [11, 19, 20]. Comparing our findings with these studies is difficult due to differences in categories of ICVHM, the level of confounder adjustment, duration of follow-up, and definition of CKD event. Rebholz et al. in the context of the ARIC study, found in 14832 subjects with a mean (SD) age of 54.7 (5.5) years, compared to 0 ideal health factors, having five or ≥ 6 health factors were associated with 73% and 81% lower risk of CKD. [11] Hou et al. in China found that among all age groups (< 40, 40–59, and ≥ 60 years), each one score increase in ICVHM per year was associated

with 11% (0.89, 0.87–0.91) reduction of CKD risk; however, the authors did not consider eGFR as an important confounder in their analysis [11, 19, 20]. Another study assessed the association of ICVHM and ESRD; it showed a graded association between a higher score of ICVHM and a lower incidence rate of ESRD, as those with 4 ICVHM had about 50% lower risk for the event compared to the reference category. However, this association was no longer significant after adjusting for eGFR, while our significant HRs persisted [11, 19, 20].

Almost all of the components of ICVHM have associated with CKD development. Individual-level data analysis conducted on 34 multinational cohorts from the CKD Prognosis Consortium, including 5 222 711 subjects from 28 countries, indicated that among diabetic and non-diabetic individuals, hypertension, having higher BMI, and ever smoker were significant predictors of CKD development [32]. Similar to our study, previous studies have also found that being in the healthier categories of BMI, BP and FPG was associated with a lower risk of CKD [11, 19, 20]. The results derived from the ARIC study showed that being in the intermediate and ideal categories of BMI was associated with 20% and 26% reduced risk of CKD, respectively. The corresponding values for BP were 27% and 50%, and for FPG, were 60% and 63% [11, 19, 20]. Contrary to our results, Hou et al. and Rebholz et al. found significant associations between smoking and low PA with an increased risk of developing CKD [11, 19]. In the current study, through different components of ICVHM, we did not find the harmful risk of smoking and low PA on incident low eGFR. A recent meta-analysis of prospective cohort studies showed that being a current smoker was associated with about 27% higher risk of incident CKD; however, there was significant heterogeneity between included studies ($I^2 \approx 80\%$) [33, 34]. Regarding the relationship of PA on incident CKD, a meta-analysis by Zhu et al. found that the highest vs. the lowest PA level was associated with reduced odds of CKD with high heterogeneity between the included studies ($I^2 \approx 84\%$). Furthermore, the authors also demonstrated that every 10 MET h/week increment of PA was associated with a reduction of 2% in CKD risk [35].

In a subgroup population with dietary data, we did not find the beneficial impact of having intermediate/ideal categories of nutrition status compared to its poor one on incident low eGFR. In the Chinese population, there were associations between worse ICVHM and ESRD, CKD and diabetes incidence; however, in these studies, only the information on salt intake was used, and there was no detailed information on dietary habits [19, 36, 37]. Previous studies on the TLGS population showed that diabetes risk reduction score which was based on eight dietary components (cereal fiber, nuts, coffee, polyunsaturated fat to saturated fat ratio, glycemic index, sugar-sweetened beverages, trans fatty acids, red and processed meat) was associated with 33% less likely to have CKD in the highest versus the lowest quartile of this score [38]. A dietary approach to stop hypertension diet and the Mediterranean diet (rich in whole grain, fruit and vegetables) have been found to reduce CKD risk by 59 and 47%, respectively [39]. Consumption of sugar-sweetened beverages ≥ 4 servings/week has been reported to increase the risk of CKD, compared to 0.5 servings/week [40]. High dietary sodium (7.84 gr) increased the risk of CKD incidence by 64% due to harmful effects on renal function [41]. Regarding the above-mentioned issues, it is better to score indices based on the recommended amounts of local food groups than the amounts based on different populations to estimate the risk of CKD effectively.

As strength, the present study was conducted in the context of the largest population based cohort study of the MENA region with a high burden of CKD. In this study, the association between ICVHM and the risk of incident low eGFR was assessed using both continuous and categorical approaches. Moreover, we also considered eGFR as an important confounder/mediator in our data analysis.

There are some limitations in our study. First; although the most epidemiological and interventional studies on CKD use single serum creatinine measurement, we considered low eGFR as the outcome instead of CKD due to lacking data on proteinuria. Second, we defined the outcome based on a single GFR estimation which has a high degree of intra-individual variability. However, consecutive GFR estimations would not extenuate the association of the predictors with the outcome. Third, we did not calibrate our serum creatinine measurement to the Cleveland Clinic, nor did we validate the eGFR formula (CKD-EPI) in a local population. This could also affect the incidence of low eGFR. Forth, diet information was not available for all participants; however, sensitivity analysis showed that dietary habits were not significantly associated with the risk of incident low eGFR. Fifth, some medications, such as Ibesartan, has some favorable impact in the course of renal failure among normotensive T2DM patients and reduced albumin excretion rate [42]; however, in our study out of 162 hypertension medication user, only 82 individuals were on Angiotensin receptor blockers. Moreover, this medication entered Iranian pharmacopeia in 2019.

## Conclusion

Based on the findings of this study, the ICVHM score was associated with a reduced risk of developing low eGFR in the middle-aged population during more than a decade follow-up. Among the various components of ICVHM, being in the ideal categories of BMI, FPG, and BP had a significant protective role against low eGFR incidence. Attainment of ideal cardiovascular health may potentially have substantial benefits for preventing CKD among the non-elder Iranian population.

## Supporting information

**S1 File.**
(DOCX)

**S1 Data.**
(XLSX)

## Acknowledgments

The authors wish to thank the study participants and the TLGS research team for their passionate support. This article has been extracted from the fellowship thesis of Dr Fatemeh Alizadeh.

## Author Contributions

**Conceptualization:** Maryam Tohidi, Fereidoun Azizi, Farzad Hadaegh.

**Formal analysis:** Mitra Hasheminia, Firoozeh Hosseini-Esfahani.

**Methodology:** Fatemeh Alizadeh, Maryam Tohidi, Farzad Hadaegh.

**Supervision:** Maryam Tohidi, Fereidoun Azizi, Farzad Hadaegh.

**Writing – original draft:** Fatemeh Alizadeh.

**Writing – review & editing:** Fatemeh Alizadeh, Maryam Tohidi, Firoozeh Hosseini-Esfahani, Farzad Hadaegh.

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
