## [Decision Letter · Decision Letter 0]

18 Jul 2022

PONE-D-22-17476Association of Ideal Cardiovascular Health Metrics with Incident Chronic Kidney Disease: more than a decade follow-up in the Tehran Lipid and Glucose Study (TLGS)PLOS ONE

Dear Dr. tohidi,

Thank you for submitting your manuscript to PLOS ONE. After careful consideration, we feel that it has merit but does not fully meet PLOS ONE’s publication criteria as it currently stands. Therefore, we invite you to submit a revised version of the manuscript that addresses the points raised during the review process.

ACADEMIC EDITOR:Please submit your revised manuscript by two weeks. If you will need more time than this to complete your revisions, please reply to this message or contact the journal office at plosone@plos.org. Please include the following items when submitting your revised manuscript:A rebuttal letter that responds to each point raised by the academic editor and reviewer(s). You should upload this letter as a separate file labeled 'Response to Reviewers'.A marked-up copy of your manuscript that highlights changes made to the original version. You should upload this as a separate file labeled 'Revised Manuscript with Track Changes'.An unmarked version of your revised paper without tracked changes. You should upload this as a separate file labeled 'Manuscript'.

We look forward to receiving your revised manuscript.

Kind regards,

Ferdinando Carlo Sasso, PhD, MD

Academic Editor

PLOS ONE

Journal Requirements:

2. We note that your study is based on the clinical trials described in the references 20 and 21. Please provide the clinical trial registration numbers for these trials in your manuscript.

"This study was initially supported in part by grant No. 121 from the National Research Council of the Islamic Republic of Iran and then by grant No 30382 from the Shahid Beheshti University of Medical Sciences."

Reviewers' comments:

Reviewer's Responses to Questions

**Comments to the Author**

1. Is the manuscript technically sound, and do the data support the conclusions?

Reviewer #1: Yes

Reviewer #2: Partly

2. Has the statistical analysis been performed appropriately and rigorously? 

Reviewer #1: Yes

Reviewer #2: I Don't Know

3. Have the authors made all data underlying the findings in their manuscript fully available?

Reviewer #1: Yes

Reviewer #2: Yes

4. Is the manuscript presented in an intelligible fashion and written in standard English?

Reviewer #1: Yes

Reviewer #2: Yes

5. Review Comments to the Author

Reviewer #1: Alizadeh and colleagues carried out a prospective study examining the association between ideal cardiovascular health metrics and incident chronic kidney disease. In a cohort of 7388 patients, they found that ideal and intermediate categories of BMI, BP and ideal categories of FBP are associated with lower risk of incident CKD in middle-aged subjects while only intermediate and ideal categories of BP associated with lower CKD risk.

This manuscript could be of some interest, but I have several concerns. Briefly, my comments:

1) In the Abstract the following sentence needs to be rephrased because there are not only HR but also the percentage of risk reduction “the corresponding HRs (95% confidence intervals) were 13% (0.87, 0.77-0.99), 16% (0.84, 0.76-0.99), 21% (0.79, 0.68-0.93), 30% (0.70, 0.60-0.83) and 24% (0.76, 0.64-0.91)”. Furthermore, in the conclusion of abstract Authors stated that Results were found only in middle-aged while there are some findings also in elderly subjects. Abstract needs to be improved

2) In Methods Section, Authors indicated that at baseline in TLGS were excluded subjects without CKD, but definition of this disease is not only based on GFR but also urinary anomalies such as proteinuria and albuminuria. Could Authors exclude that, at baseline, patients had a significant proteinuria? This is a crucial point in the evaluation of incident CKD. If data on proteinuria were not available, I suggest referring to low eGFR (<60 ml/min) as outcome instead of CKD.

3) Creatinine levels were assessed by kinetic colorimetric Jaffè and not with enzymatic assay, but CKD-EPI formula was used to estimate GFR. The Authors did not perform a correction on creatinine values to use CKD-EPI formula. This could limit GFR evaluation. I suggest that Since creatinine was not standardized to isotope dilution mass spectrometry values, Authors could reduce creatinine values by 5% as indicated by Skali et al. and, after, could use CKD-EPI formula (PMID: 21884875)

4) Surprisingly, 68.3% of elderly subjects (>65 y) enrolled in the study developed incident CKD. I think that a selection bias could be the reason for this high incidence of CKD. Probably, Authors could limit the analyses on middle-aged population.

Reviewer #2: I’ve read with interest the draft “Association of ideal cardiovascular health metrics with incident chronic 2 kidney disease: more than a decade follow-up in the Tehran Lipid and 3 Glucose Study (TLGS)”, by Fatemeh Alizadeh, Maryam Tohidi, Mitra Hasheminia, Firoozeh Hosseini-Esfahani, Fereidoun Azizi, Farzad Hadaegh

However, some important issues have to be raised.

Introduction:

- Line 46-47. Authors should briefly introduce and define the main non-communicable diseases (NCDs).

- Line 65-66. A multifactorial treatment aimed at multiple risk factors reduces the risk of cardiovascular events in populations with type 2 diabetes mellitus and renal impairment. This issue is addressed in some recent literature which should be added to the references. Gaede P et al, N Engl J Med. 2003;348(5):383-393. doi:10.1056/NEJMoa021778; Sasso FC et al Cardiovasc Diabetol. 2021;20(1):145. Published 2021 Jul 16. doi:10.1186/s12933-021-01343-1

Methods and Materials:

- Line 128-129. Font typing error.

Discussion:

- Line 251. “china” is a typing error.

- Line 237-239. “Among middle-aged participants, the intermediate and ideal categories of BMI and BP, and the ideal category of FPG and in elderly individuals, only intermediate and ideal BP was associated with reduced risk of CKD”. The authors should better clarify the pathophysiological mechanisms/hypotheses supporting this sentence.

- Line 262. “Actually, no association was demonstrated between ICVHM and incident CKD among elder population”. It therefore seems that ICVHM does not affect the prognosis of elderly patients. However, some recent literature seems in contrast with this conclusion. The authors should better clarify the pathophysiological mechanisms/hypotheses supporting this sentence.

6. PLOS authors have the option to publish the peer review history of their article (what does this mean?). If published, this will include your full peer review and any attached files.

Reviewer #1: No

Reviewer #2: No

---

## [Author Response · Author response to Decision Letter 0]

5 Nov 2022

Journal Requirements:

Response: Thank you, done.

2. We note that your study is based on the clinical trials described in the references 20 and 21. Please provide the clinical trial registration numbers for these trials in your manuscript.

Response: Agreed, done, the trial registration number (ISRCTN52588395) was added to the text. ” (page, line: 89)

Response: Agreed, corrected. "This study was initially supported in part by grant No. 121 from the National Research Council of the Islamic Republic of Iran and then by grant No 30382 from the Shahid Beheshti University of Medical Sciences." ” (page, line: 100-103)

Response: The authors declare that they have no competing interests.

Response: Agreed, the ethic statement and related approval number was mentioned in the Method section. ” (page, line: 100-103)

Reviewers' comments:

Reviewer's Responses to Questions

Comments to the Author

1. Is the manuscript technically sound, and do the data support the conclusions?

Reviewer #1: Yes

Reviewer #2: Partly

2. Has the statistical analysis been performed appropriately and rigorously? 

Reviewer #1: Yes

Reviewer #2: I Don't Know

3. Have the authors made all data underlying the findings in their manuscript fully available?

Reviewer #1: Yes

Reviewer #2: Yes

4. Is the manuscript presented in an intelligible fashion and written in standard English?

Reviewer #1: Yes

Reviewer #2: Yes

5. Review Comments to the Author

Reviewer #1: Alizadeh and colleagues carried out a prospective study examining the association between ideal cardiovascular health metrics and incident chronic kidney disease. In a cohort of 7388 patients, they found that ideal and intermediate categories of BMI, BP and ideal categories of FBP are associated with lower risk of incident CKD in middle-aged subjects while only intermediate and ideal categories of BP associated with lower CKD risk.

This manuscript could be of some interest, but I have several concerns. Briefly, my comments:

1) In the Abstract the following sentence needs to be rephrased because there are not only HR but also the percentage of risk reduction “the corresponding HRs (95% confidence intervals) were 13% (0.87, 0.77-0.99), 16% (0.84, 0.76-0.99), 21% (0.79, 0.68-0.93), 30% (0.70, 0.60-0.83) and 24% (0.76, 0.64-0.91)”. Furthermore, in the conclusion of abstract Authors stated that Results were found only in middle-aged while there are some findings also in elderly subjects. Abstract needs to be improved

Response: The mentioned part of the abstract was rephrased as below: “the corresponding HRs (95% confidence intervals) were (0.87, 0.77-0.99), (0.84, 0.76-0.99), (0.79, 0.68-0.93), (0.70, 0.60-0.83) and (0.76, 0.64-0.91).” (page , line: 37-39)

2) In Methods Section, Authors indicated that at baseline in TLGS were excluded subjects without CKD, but definition of this disease is not only based on GFR but also urinary anomalies such as proteinuria and albuminuria. Could Authors exclude that, at baseline, patients had a significant proteinuria? This is a crucial point in the evaluation of incident CKD. If data on proteinuria were not available, I suggest referring to low eGFR (<60 ml/min) as outcome instead of CKD.

Response: Thank you for this valuable comment. Since, in this cohort study, assay for urine protein was not performed at baseline and follow-up and CKD was defined based on eGFR, we corrected the manuscript and Tables according to this comment.” (page , line: The whole text)

3) Creatinine levels were assessed by kinetic colorimetric Jaffè and not with enzymatic assay, but CKD-EPI formula was used to estimate GFR. The Authors did not perform a correction on creatinine values to use CKD-EPI formula. This could limit GFR evaluation. I suggest that Since creatinine was not standardized to isotope dilution mass spectrometry values, Authors could reduce creatinine values by 5% as indicated by Skali et al. and, after, could use CKD-EPI formula (PMID: 21884875)

Response: Agreed, this issue for correcting the creatinine levels that has been considered in the data analysis, was added to the text. ” (page, line: 135-137)

4) Surprisingly, 68.3% of elderly subjects (>65 y) enrolled in the study developed incident CKD. I think that a selection bias could be the reason for this high incidence of CKD. Probably, Authors could limit the analyses on middle-aged population.

Response: Agreed, All information related to age over 65 years was removed from the draft article.

Reviewer #2: I’ve read with interest the draft “Association of ideal cardiovascular health metrics with incident chronic 2 kidney disease: more than a decade follow-up in the Tehran Lipid and 3 Glucose Study (TLGS)”, by Fatemeh Alizadeh, Maryam Tohidi, Mitra Hasheminia, Firoozeh Hosseini-Esfahani, Fereidoun Azizi, Farzad Hadaegh

However, some important issues have to be raised.

Introduction :

- Line 46-47. Authors should briefly introduce and define the main non-communicable diseases (NCDs).

Response: The mentioned items were corrected. ” (page, line: 46-49)

- Line 65-66. A multifactorial treatment aimed at multiple risk factors reduces the risk of cardiovascular events in populations with type 2 diabetes mellitus and renal impairment. This issue is addressed in some recent literature which should be added to the references. Gaede P et al, N Engl J Med. 2003;348(5):383-393. doi:10.1056/NEJMoa021778; Sasso FC et al Cardiovasc Diabetol. 2021;20(1):145. Published 2021 Jul 16. doi:10.1186/s12933-021-01343-1

Response: Thank you, for suggesting these references. The suggested references were added to the text. ” (page, line: 66-69)

Methods and Materials:

- Line 128-129. Font typing error.

Response: Thank you, the mentioned items were corrected. ” (page, line: 134-135)

Discussion:

- Line 251. “china” is a typing error.

Response: Thank you, the mentioned items were corrected. ” (page, line: 238)

- Line 237-239. “Among middle-aged participants, the intermediate and ideal categories of BMI and BP, and the ideal category of FPG and in elderly individuals, only intermediate and ideal BP was associated with reduced risk of CKD”. The authors should better clarify the pathophysiological mechanisms/hypotheses supporting this sentence.

Response: Thank you, all information related to age over 65 years was removed from the draft article according to the opinions of reviewers.

- Line 262. “Actually, no association was demonstrated between ICVHM and incident CKD among elder population”. It therefore seems that ICVHM does not affect the prognosis of elderly patients. However, some recent literature seems in contrast with this conclusion. The authors should better clarify the pathophysiological mechanisms/hypotheses supporting this sentence.

Response: agreed, according to comment number 4 of the first reviewer and the last comment of the second reviewer, we decided to remove participant over 65 years old from the draft of the article.

6. PLOS authors have the option to publish the peer review history of their article (what does this mean?). If published, this will include your full peer review and any attached files.

Do you want your identity to be public for this peer review? For information about this choice, including consent withdrawal, please see our Privacy Policy.

Reviewer #1: No

Reviewer #2: No

---

## [Decision Letter · Decision Letter 1]

21 Nov 2022

PONE-D-22-17476R1Association of ideal cardiovascular health metrics with incident low estimated glomerular filtration rate: more than a decade follow-up in the Tehran Lipid and Glucose Study (TLGS)PLOS ONE

Dear Dr. tohidi,

Thank you for submitting your manuscript to PLOS ONE. After careful consideration, we feel that it has merit but does not fully meet PLOS ONE’s publication criteria as it currently stands. Therefore, we invite you to submit a revised version of the manuscript that addresses the points raised during the review process.

ACADEMIC EDITOR:In particular, the authors have to address the issues raised by the third reviewer.

We look forward to receiving your revised manuscript.

Kind regards,

Ferdinando Carlo Sasso, PhD, MD

Academic Editor

PLOS ONE

Additional Editor Comments (if provided):

Please, address the issues raised by reviewer 2 and 3.

Reviewers' comments:

Reviewer's Responses to Questions

**Comments to the Author**

1. If the authors have adequately addressed your comments raised in a previous round of review and you feel that this manuscript is now acceptable for publication, you may indicate that here to bypass the “Comments to the Author” section, enter your conflict of interest statement in the “Confidential to Editor” section, and submit your "Accept" recommendation.

Reviewer #1: All comments have been addressed

Reviewer #2: (No Response)

Reviewer #3: (No Response)

2. Is the manuscript technically sound, and do the data support the conclusions?

Reviewer #1: Yes

Reviewer #2: Yes

Reviewer #3: No

3. Has the statistical analysis been performed appropriately and rigorously? 

Reviewer #1: Yes

Reviewer #2: Yes

Reviewer #3: No

4. Have the authors made all data underlying the findings in their manuscript fully available?

Reviewer #1: Yes

Reviewer #2: Yes

Reviewer #3: No

5. Is the manuscript presented in an intelligible fashion and written in standard English?

Reviewer #1: Yes

Reviewer #2: Yes

Reviewer #3: No

6. Review Comments to the Author

Reviewer #1: All the criticisms raised were solved by the Authors. I do not have any further comment on the manuscript.

Reviewer #2: Dear Editor,

I’ve re revised the paper “Association of ideal cardiovascular health metrics with incident low estimated

glomerular filtration rate: more than a decade follow-up in the Tehran Lipid and

Glucose Study (TLGS)”, by Fatemeh Alizadeh et al.

However, some important issues need to be raised.

- Some drugs seem to have a beneficial effect on patients with chronic kidney disease. Please, briefly discuss this point by adding the reference (Irbesartan reduces the albumin excretion rate in microalbuminuric type 2 diabetic patients independently of hypertension: A randomized double-blind placebo-controlled crossover study. DOI 10.2337/diacare.25.11.19)

- The role of drugs as potential cofounders was not discussed. Please, briefly discuss this important issue.

Reviewer #3: PONE-D-22-17476R1: statistical review

SUMMARY. This is a cohort study to evaluate the association between ideal cardiovascular health metrics (ICVHM) and time up an event of low estimated glomerular filtration rate (eGFR). The core statistical analysis relies on a battery of Cox regression models where the effect of ICVHM is adjusted by including available confounding factors. I have several concerns about the estimation method (major issue 1), sample selection (major issue 2), model specification (major issue 3). I also append some specific points that should be addressed.

MAJOR ISSUES.

1. Lines 178-180. Although data are interval censored, event times are estimated by mid-points and treated as they were exact times. This approach is a potential source of bias and it is not correct. The authors should repeat the statistical analysis by accounting for the additional uncertainty brought by interval censoring. Cox regression analysis for interval-censored data can be easily perfromed in Stata, which is the software exploited by the authors: see the command "stintcox".

2. Lines 91-97. The statistical analysis is made on a subset of the initial sample. To avoid issues of selection bias, the authors should explain why the final sample can be considered as a random subset of the initial sample. See also specific point 1 below.

3. If I have understood well, covariates are included at baseline. However, some of this covariates (e.g. age) are time-varying and certainly have a time-varying confounding effects on the time up to the event. I'd welcome an analysis that adjusts for time-varying covariates. Alternately, the authors should explain why time-varying covariates are not appropriate here.

SPECIFIC POINTS

1. In lines 77-81, the authors talk about a large population-based study conducted in the Middle East and North Africa (MENA) during more than 15 years of follow-up. In lines 84-89, they instead talk about a 3-year study in Teheran, which seems the study considered in this paper. What is the link between the two studies? Why was the large population 15-year study evoked, if the actual study of interest is the 3-year study? Please clarify.

2. Lines 182-184. I'm glad that the authors have made some diagnostics of the model. However, the results are declared but not displayed. These secondary results should be commented and displayed as supplementary material.

3. Data. It is good that all the data are available without restrictions in a repository. However, is this repository public? If it is not, then the authors should provide precise instructions for obtaining the data.

4. There is a tendency to interpret the results as they were the outcome of a logistic regression model where the dependent variable is a binary variable that records the occurrence of the event. However, the dependent variable in a Cox model is the time up to the event and covariates modulate the hazard function. The text should be modified accordingly.

5. There some typos across the text, please check.

7. PLOS authors have the option to publish the peer review history of their article (what does this mean?). If published, this will include your full peer review and any attached files.

Reviewer #1: No

Reviewer #2: No

Reviewer #3: No

---

## [Author Response · Author response to Decision Letter 1]

5 Feb 2023

Reviewer #1: All the criticisms raised were solved by the Authors. I do not have any further comment on the manuscript.

Response: Thank you for the positive comment.

Reviewer #2: 

Dear Editor,

I’ve re revised the paper “Association of ideal cardiovascular health metrics with incident low estimated

glomerular filtration rate: more than a decade follow-up in the Tehran Lipid and

Glucose Study (TLGS)”, by Fatemeh Alizadeh et al.

However, some important issues need to be raised.

- Some drugs seem to have a beneficial effect on patients with chronic kidney disease. Please, briefly discuss this point by adding the reference (Irbesartan reduces the albumin excretion rate in microalbuminuric type 2 diabetic patients independently of hypertension: A randomized double-blind placebo-controlled crossover study. DOI 10.2337/diacare.25.11.19)

Response: Agreed, we used this important study in the discussion section. (Line: 331-335)

- The role of drugs as potential cofounders was not discussed. Please, briefly discuss this important issue.

Response: Agreed, we discussed this important issue in the limitation part of the manuscript. (Line: 331-335)

Reviewer #3: PONE-D-22-17476R1: statistical review

SUMMARY. This is a cohort study to evaluate the association between ideal cardiovascular health metrics (ICVHM) and time up an event of low estimated glomerular filtration rate (eGFR). The core statistical analysis relies on a battery of Cox regression models where the effect of ICVHM is adjusted by including available confounding factors. I have several concerns about the estimation method (major issue 1), sample selection (major issue 2), model specification (major issue 3). I also append some specific points that should be addressed.

MAJOR ISSUES.

1. Lines 178-180. Although data are interval censored, event times are estimated by mid-points and treated as they were exact times. This approach is a potential source of bias and it is not correct. The authors should repeat the statistical analysis by accounting for the additional uncertainty brought by interval censoring. Cox regression analysis for interval-censored data can be easily performed in Stata, which is the software exploited by the authors: see the command "stintcox".

Response: The theory for the analysis of interval-censored data has been developed over the past four decades. The midpoint of the censoring interval is a common practice in analyzing interval-censored survival data in medical and reliability studies to simplify the interval censoring structure and then apply standard analytic techniques for multivariate survival data [1-4]. The availability of software for right censoring might well be the main reason for this simplifying practice [5]. Although the performance of the mid-time approach may lead to biased estimates because of limited information due to interval-censoring (the event might happen at the beginning or end of the follow-up duration) Cox analysis has minimal impact from the differing assumptions about the timing of events [6]. Furthermore, our research team just published several papers using a similar approach in high-impact journals, including the Plos One journal [7-9]. 

Despite this, as requested by the reviewer, to test the robustness of our findings, we performed Cox regression analysis with interval-censoring using the "stintcox" command in STATA as a sensitivity analysis. Accordingly, the results using the interval-censored approach did not differ from those that obtained from the mid-point approach. (Lines 204-206 and 252-254)

2. Lines 91-97. The statistical analysis is made on a subset of the initial sample. To avoid issues of selection bias, the authors should explain why the final sample can be considered as a random subset of the initial sample. See also specific point 1 below.

Response: Agreed, we more clarified the method section of the revised manuscript. Moreover, to address the selection bias [10], propensity score (PS), the estimated probability that a participant would have followed in the study, was computed using maximum likelihood logistic regression analysis among the study population. The multivariable analysis, including PS as another covariate, was performed as another sensitivity analysis; accordingly, the findings were essentially unchanged. So the issue of selection bias might not affect our data analysis. (Lines: 196-199 and 249-252)

3. If I have understood well, covariates are included at baseline. However, some of this covariates (e.g. age) are time-varying and certainly have a time-varying confounding effects on the time up to the event. I'd welcome an analysis that adjusts for time-varying covariates. Alternately, the authors should explain why time-varying covariates are not appropriate here.

Response: Agreed, to test the robustness of our findings, we treated covariates, including age and marital status, as time-varying confounders in the sensitivity analysis; our results remained essentially unchanged, as shown in the supplementary material. (Lines: 205-207 and 254-256)

SPECIFIC POINTS

1. In lines 77-81, the authors talk about a large population-based study conducted in the Middle East and North Africa (MENA) during more than 15 years of follow-up. In lines 84-89, they instead talk about a 3-year study in Teheran, which seems the study considered in this paper. What is the link between the two studies? Why was the large population 15-year study evoked, if the actual study of interest is the 3-year study? Please clarify.

Response: The TLGS study is a prospective cohort study started in 1999 and is planned to be continued for about 20 years for the non-communicable outcomes. The participant are follow up every 3 years (phase 1: 1999-2002, phase 2: 2002-2005, phase 3: 2005-2008, phase 4: 2008-2011, phase 5: 2011-2014, phase 6: 2014-2018, phase 7: 2018-2021) Detailed information regarding the TLGS is available elsewhere (trial registration number: ISRCTN52588395). The current study includes those who participated in phase 3 of TLGS (2005-2008) and followed till February 2021; the median follow-up time for the current study was 15 years. More description was added to the method section to clarify this issue [11, 12]. (Line: 96-97)

2. Lines 182-184. I'm glad that the authors have made some diagnostics of the model. However, the results are declared but not displayed. These secondary results should be commented and displayed as supplementary material.

Response: All P values for proportionality assumptions were > 0.1 in different multivariable-adjusted models. (Line: 193-194) and (Supplementary Tables S1 and S2)

3. Data. It is good that all the data are available without restrictions in a repository. However, is this repository public? If it is not, then the authors should provide precise instructions for obtaining the data.

Response: The TLGS data is not public, however, Azizi F. is the pioneer researcher of the TLGS, and other authors of this manuscript are among the research team of TLGS in the research institute for endocrine sciences (RIES), so we have asses to the TLGS data.

4. There is a tendency to interpret the results as they were the outcome of a logistic regression model where the dependent variable is a binary variable that records the occurrence of the event. However, the dependent variable in a Cox model is the time up to the event and covariates modulate the hazard function. The text should be modified accordingly.

Response: Agreed, we modified the discussion section accordingly. (Line: 259-267)

5. There some typos across the text, please check.

Response: Done.

1. Kim MY, Xue X. The analysis of multivariate interval‐censored survival data. Statistics in Medicine. 2002;21(23):3715-26.

2. Rücker G, Messerer D. Remission duration: an example of interval‐censored observations. Statistics in Medicine. 1988;7(11):1139-45.

3. Lin DY, Wei L-J. The robust inference for the Cox proportional hazards model. Journal of the American statistical Association. 1989;84(408):1074-8.

4. Odell PM, Anderson KM, D'Agostino RB. Maximum likelihood estimation for interval-censored data using a Weibull-based accelerated failure time model. Biometrics. 1992:951-9.

5. Gómez G, Calle ML, Oller R, Langohr K. Tutorial on methods for interval-censored data and their implementation in R. Statistical Modelling. 2009;9(4):259-97.

6. Singh RS, Totawattage DP. The statistical analysis of interval-censored failure time data with applications. 2013.

7. Derakhshan A, Sardarinia M, Khalili D, Momenan AA, Azizi F, Hadaegh F. Sex specific incidence rates of type 2 diabetes and its risk factors over 9 years of follow-up: Tehran Lipid and Glucose Study. PloS one. 2014;9(7):e102563.

8. Ramezankhani A, Azizi F, Hadaegh F. Associations of marital status with diabetes, hypertension, cardiovascular disease and all-cause mortality: A long term follow-up study. PloS one. 2019;14(4):e0215593.

9. Derakhshan A, Bagherzadeh‐Khiabani F, Arshi B, Ramezankhani A, Azizi F, Hadaegh F. Different combinations of glucose tolerance and blood pressure status and incident diabetes, hypertension, and chronic kidney disease. Journal of the American Heart Association. 2016;5(8):e003917.

10. Rubin DB. Estimating causal effects from large data sets using propensity scores. Annals of internal medicine. 1997;127(8_Part_2):757-63.

11. Azizi F, Rahmani M, Emami H, Mirmiran P, Hajipour R, Madjid M, et al. Cardiovascular risk factors in an Iranian urban population: Tehran lipid and glucose study (phase 1). Sozial-und präventivmedizin. 2002;47(6):408-26. doi: 10.1007/s000380200008.

12. Azizi F, Ghanbarian A, Momenan AA, Hadaegh F, Mirmiran P, Hedayati M, et al. Prevention of non-communicable disease in a population in nutrition transition: Tehran Lipid and Glucose Study phase II. Trials. 2009;10(1):1-15. doi: 10.1186/1745-6215-10-5.

---

## [Decision Letter · Decision Letter 2]

23 Feb 2023

Association of ideal cardiovascular health metrics with incident low estimated glomerular filtration rate: more than a decade follow-up in the Tehran Lipid and Glucose Study (TLGS)

PONE-D-22-17476R2

Dear Dr. tohidi,

We’re pleased to inform you that your manuscript has been judged scientifically suitable for publication and will be formally accepted for publication once it meets all outstanding technical requirements.

Kind regards,

Ferdinando Carlo Sasso, PhD, MD

Academic Editor

PLOS ONE

Additional Editor Comments (optional):

No further comments

Reviewers' comments:

Reviewer's Responses to Questions

**Comments to the Author**

1. If the authors have adequately addressed your comments raised in a previous round of review and you feel that this manuscript is now acceptable for publication, you may indicate that here to bypass the “Comments to the Author” section, enter your conflict of interest statement in the “Confidential to Editor” section, and submit your "Accept" recommendation.

Reviewer #1: All comments have been addressed

Reviewer #2: All comments have been addressed

Reviewer #3: All comments have been addressed

2. Is the manuscript technically sound, and do the data support the conclusions?

Reviewer #1: Yes

Reviewer #2: Yes

Reviewer #3: (No Response)

3. Has the statistical analysis been performed appropriately and rigorously? 

Reviewer #1: Yes

Reviewer #2: Yes

Reviewer #3: (No Response)

4. Have the authors made all data underlying the findings in their manuscript fully available?

Reviewer #1: Yes

Reviewer #2: Yes

Reviewer #3: (No Response)

5. Is the manuscript presented in an intelligible fashion and written in standard English?

Reviewer #1: Yes

Reviewer #2: Yes

Reviewer #3: (No Response)

6. Review Comments to the Author

Reviewer #1: All the criticisms raised were solved by the Authors. I have no further comment to do on the manuscript.

Reviewer #2: (No Response)

Reviewer #3: (No Response)

7. PLOS authors have the option to publish the peer review history of their article (what does this mean?). If published, this will include your full peer review and any attached files.

Reviewer #1: No

Reviewer #2: No

Reviewer #3: No

---

## [Editor Report · Acceptance letter]

27 Feb 2023

PONE-D-22-17476R2 

Association of ideal cardiovascular health metrics with incident low estimated glomerular filtration rate: more than a decade follow-up in the Tehran Lipid and Glucose Study (TLGS) 

Dear Dr. tohidi:

I'm pleased to inform you that your manuscript has been deemed suitable for publication in PLOS ONE. Congratulations! Your manuscript is now with our production department. 

Kind regards, 

on behalf of

Professor Ferdinando Carlo Sasso 

Academic Editor

PLOS ONE